

# Biochemical and molecular responses of maize to low and high temperatures in symbiosis with mixed arbuscular mycorrhizal fungi cultures

Vedia Turudu[1], Imren Kutlu[2] and Nurdilek Gulmezoglu[3]

[1] Republic of Turkey Ministry of Agriculture and Forestry General Directorate of Agricultural Research and Policies, Ankara, Turkey
[2] Field Crops, Osmangazi University, Eskisehir, Turkey
[3] Department of Plant Nutrition and Soil Science, Faculty of Agriculture, Eskisehir Osmangazi University, Eskihehir, Turkey

Corresponding author
Imren Kutlu, ikutlu@ogu.edu.tr

## ABSTRACT

In this study, changes in mineral element concentrations, physiological parameters, and gene expression of heat shock proteins were investigated in maize plants subjected to mycorrhiza under low and high temperature stress. The application of seven different temperatures (5 °C, 10 °C, 15 °C, 25 °C, 35 °C, 40 °C, and 45 °C) under five different mixed arbuscular mycorrhizal fungi (AMF) culture treatments (M0, M1, M2, M3, and M4) constituted the factors of the experiment. With the application of mycorrhiza, the plant dry weight was found to be the highest at 25 °C, and the M3 group was applied. The highest values in mineral element concentrations were detected at 25 °C in the maize plant, where M4 had N, P, K, Ca, and Fe concentrations; M3 had Cu and Mn concentrations; and M2 had Mg and Zn concentrations. Lipid peroxidation gradually increased with temperature changes in all the applications, and the protective effect of proline was more pronounced at high temperatures than at low temperatures. Antioxidant enzyme activities were altered by applications of mycorrhiza and temperature. For all mycorrhiza applications, the expression of HSP70 and HSP90 reached a maximum at 10 °C, 40 °C, and 45 °C. It has been revealed that low- and high-temperature applications in maize plants cause serious changes in the mycorrhizal symbiosis on the basis of investigated parameters, and these changes occur at different levels depending on the temperature changes and the differences between mixed AMF cultures. However, it can be said that the M3 application has the capacity to facilitate the growth of maize even in conditions of low (−10 °C) and high (45 °C) temperature.

## INTRODUCTION

Maize (*Zea mays* L.) is a crop cultivated across the globe for human consumption, industrial application, and animal feed. There is an urgent need to enhance yields without expanding planting areas, as the demand for maize around the world keeps growing.

However, substantial temperature fluctuations beyond the optimal range of 25–30 °C are increasingly impacting critical growth stages, rendering climate change a substantial threat to maize production (*Ahmad et al., 2024*). Maize seedlings are sensitive to low temperatures, especially during early development. Injury often occurs when temperatures drop to 8 °C in early spring (*Bhattacharya, 2022*). Maize, as a secondary crop, is also susceptible to heat stress when the temperature reaches 35 °C during seedling development (*Chandra et al., 2023*).

When maize plants are exposed to extreme temperatures, one of the earliest responses occurs at the level of lipid composition and metabolic processes, leading to alterations in cell membrane integrity. Membrane destabilization results in ion leakage, enhanced production of reactive oxygen species (ROS), and lipid peroxidation (*Waqas et al., 2021*). Excessive ROS accumulation subsequently damages lipids, proteins, nucleic acids, and other essential macromolecules. To mitigate these deleterious effects, plants have evolved sophisticated defense strategies that involve both enzymatic and non-enzymatic antioxidant systems. In addition, osmotic adjustments, such as the accumulation of proline and soluble sugars, play an important role in stabilizing membranes and alleviating heat-induced damage (*Wang et al., 2024*).

The utilization of arbuscular mycorrhizal fungi (AMF) has emerged as a novel solution that capitalizes on the beneficial relationships between the plants and microbiome. This interest is attributed to the growing recognition of the role of microbacteria in the ecosystem as carbon sinks and nutrient providers for maize plants in the context of climate change-induced temperature stress (*Tak, 2023*). The formation of AMF symbiosis is characterized by a mutualistic relationship between arbuscular mycorrhiza and plant roots. In this symbiotic interaction, the plant provides a physical space for the fungus to thrive, while the fungus in return provides the plant with essential nutrients, including nitrogen (N), phosphorus (P), sulfur (S), zinc (Zn), and copper ($Cu^{2+}$), as well as storing carbon dioxide ($CO_2$) in the soil (*Phillips, 2017*). The increase in plant mineral nutrition through mycorrhizal symbiosis increases the resistance of the host plant to abiotic stressors. A similar emphasis should be placed on other effects, such as molecular changes in the capacity of AMF symbiosis to lessen the effects of abiotic stress (*Mitra, Djebaili & Pellegrini, 2021*).

The utilization of AMF to alleviate cold stress is becoming a more feasible approach because of its ability to improve plant performance under both stressful and normal circumstances. Notably, symbiotic relationships with AMF can increase cold tolerance in plants, even though root colonization by AMF is reduced at temperatures lower than 15 °C (*Devi, Gupta & Kapoor, 2019*). The molecular mechanisms underlying this enhanced cold tolerance involve the coordinated regulation of genes responsive to cold stress, leading to a reduction in lipid peroxidation, maintenance of cell membrane integrity, enhancement of antioxidant activity, optimization of osmolyte accumulation, improvement of root hydraulic regulation, and augmentation of photosynthesis and respiration (*Zhu, Song & Xu, 2010*; *Zhu et al., 2015*; *Liu et al., 2016a*; *Li et al., 2020*). Additionally, AMF is imperative for enhancing crop resilience to heat stress, a problem that is escalating in severity due to global warming. According to *Cabral et al. (2016)*, a mixture of five AMF isolates have been
shown to modify nutrient allocation, thereby augmenting grain production in high-temperature environments by regulating nutrient dynamics. Through a holistic approach encompassing enhanced nutrient absorption, elevated antioxidant activities, and sophisticated physiological adaptations, AMF substantially fortify plant defense systems against temperature stress. The collective effect of these enhancements is a substantial promotion of increased plant development and an enhancement of various plant species' resistance to extreme temperature shocks (*Zhu et al., 2011*; *Mathur, Sharma & Jajoo, 2018*; *Mathur et al., 2021*).

In the attributed studies, *Glomus etunicatum, Glomus mosseae, Glomus tortuosum,* and *Rhizophagus irregularis=Glomus intraradices* mycorrhizal species were utilized to mitigate the temperature stress, and similar results were reported. The capacity of AMF to enhance the cold and heat tolerance of maize plants can be regarded as a species-independent characteristic. The research conducted on the subject of temperature stress in mycorrhizal maize plants has predominantly concentrated on the symbiosis with a single mycorrhizal species. It is acknowledged that individual AMF species may exhibit divergent effects on plant growth. The phenomenon of multiple invasion of a single root patch by AMFs from different genera or species has now achieved widespread acceptance within the scientific community. However, the mechanisms underpinning this co-occupancy remain to be fully elucidated. The nature of the relationship between these organisms, whether competitive, synergistic or antagonistic, remains an area of research that is yet to be fully explored (*Boyer et al., 2015*). It is acknowledged that this relationship may be subject to variation depending on the specific plant host and/or environmental conditions. However, from an ecological perspective, it is hypothesized that a mixture of different AMF species will be more efficient in occupying ecological niches for symbiosis and better adapt to environmental fluctuations. A limited number of studies have reported complementary effects of different AMF taxa on plant development (*Amir & Crossay, 2024*). The species diversity present within AMF communities suggests functional specialization, with some species increasing aboveground or underground biomass, while others improve nutrient uptake, such as K, Mg, and Zn, which, in turn, indirectly increases chlorophyll content (*Moukarzel et al., 2023*). However, it is evident that mycorrhiza enhance plant growth and increase tolerance to abiotic stress in both single and mixed cultures.

In the case of temperature stress, AMF has been investigated in studies conducted in different plants to trigger the expression of some genes and proteins that control thermomorphogenesis, antioxidant activation, and thermosensory growth and initiate the biosynthesis of plant hormones (*Tak, 2023*). However, no study has revealed a significant relationship between AMF symbiosis and heat shock proteins (HSP) activation, one of the main mechanisms that occurs in response to heat stress. The expression of genes encoding HSPs, detoxifying enzymes, osmoprotectants, ion transporters, and transcription factors is altered at the molecular level by temperature stress. HSPs are the most commonly and significantly expressed genes among those that respond to temperature stress in a variety of heat-stressed plant species. The activation of a certain signaling sequence triggers the transcription of HSPs (*Mondal et al., 2023*). As molecular chaperones, HSPs facilitate refolding and disassembly for subsequent destruction, thereby preventing the misfolding

of proteins and protein aggregation. Additionally, HSP activity is essential for the priming and development of stress tolerance (*Balasubramani et al., 2024*). HSP70 and HSP90 are two of the HSPs that are recognized as essential for optimal plant development and for coping with stressful situations. The HSP70 and HSP90 families have been suggested as possible biomarkers in a variety of organisms subjected to different stressors, such as drought, heavy metals, salt, and extreme temperatures (*Khan et al., 2021*).

In maize plants, studies have identified HSPs that confer thermotolerance under low or high temperature conditions in the absence of mycorrhizal symbiosis (*Abou-Deif et al., 2019*; *Kumar et al., 2019*; *Zhang et al., 2020*). The findings from previous studies indicate a knowledge gap concerning the regulation of HSP90 and HSP70 proteins, which play significant roles in numerous environmental stress conditions, as well as their expression in maize. This study is the first in analyzing changes in HSP gene expression in AMF-inoculated maize plants under low and high temperature stress. The impact of stress conditions was also examined by investigating the relationships between AMF colonization, antioxidant enzyme activities, and mineral element concentrations to HSP gene expressions at low and high temperatures. Furthermore, it is pioneering in its field by comparing differences of mixed AMF cultures in terms of changes in mineral element concentrations, antioxidant enzyme activities, and HSP gene expression in maize plants inoculated with different mixed AMF cultures under low and high temperature stress.

## MATERIALS AND METHODS

### Characterization of experimental soil, plant material, and AMF inoculum

The experimental soil was taken from the Ap horizon (0–30 cm) in the Eskisehir Osmangazi University experimental research area, located at 39°48′N, 30°31′E. The soil samples were examined for physical and chemical characteristics following air drying and filtering through a 2 mm mesh. The soils analyzed were of the sandy loam texture, alkaline (pH = 7.99), salt-free, calcareous, poor in organic matter (1.8%), and deficient in phosphorus (56 kg ha$^{-1}$), zinc (0.24 mg kg$^{-1}$), and manganese (5.64 mg kg$^{-1}$), whereas potassium (2,540 kg ha$^{-1}$), iron (2.81 mg kg$^{-1}$), and copper (1.73 mg kg$^{-1}$) were sufficient.

The plant material used in this study was the FAO500 maturity group dent maize cultivar PL472, which was obtained from the Polen Seed Company (Manisa, Turkey).

Four distinct mixed AMF cultures were utilized in the study. Three of these products (M1, M2, and M3) were obtained from the mycorrhiza stocks of Diyarbakır Plant Protection Research Institute. These mixed AMF species (M1, M2, and M3) have been isolated from their natural habitats in the Diyarbakır region and propagated using maize as a trap plant. The fourth was an Endo Roots Soluble-ERS brand product, which was procured from the international commercial company Bioglobal. The names of the species that comprise the AMF cultures are listed below.

M1-DYB4 (*Glomus mosseae, Glomus intraradices*)

M2-DYB17 (*Glomus intraradices, Glomus constrictum, Glomus microcarpum*)

M3-DYB22 (*Gigaspora sp., Glomus constrictum, Glomus fasciculatum*)

M4-ERS (*Glomus intraradices, Glomus aggregatum, Glomus mosseae, Glomus clarium, Glomus monosporus, Glomus deserticola, Glomus brasilianum, Glomus etunicatum, Gigaspora margarita*)

## Sterilization of plant growth media, pots, and seeds

The growing media comprised a 2:2:1:1 ratio of soil, peat, perlite, and sand. The mixture was autoclaved at 121 °C for 2 h at a pressure of 1.5 atm for sterilization. The 20 cm height and 20 cm diameter plastic pots used in the experiments were thoroughly cleaned before the experiment by washing with tap water, followed by washing with a 10% sodium hypochlorite (NaOCl) solution, and subsequently rinsed with pure water three times. They were then sterilized with ethanol once more before planting.

The maize seeds used were soaked in 70% ethanol for 1 min and 2% sodium hypochlorite for 3 min, rinsed 4 times with distilled water, and then placed in 100 mL flasks with water and left for 1 day.

## Experimental setup, mycorrhiza applications, and plant cultivation

The experiment was set up according to a factorial experimental design, with three replications. The maize variety PL472 was cultivated under five mycorrhiza treatments (M0: without mycorrhiza/control, M1, M2, M3, and M4 mixed AMF cultures) and seven temperature treatments (5 °C, 10 °C, 15 °C, 25 °C, 35 °C, 40 °C, and 45 °C).

The sterilized experimental soil was combined with fertilizer solutions containing the necessary nutrients. These nutrients were prepared from the following substances: $(NH_4)_2SO_4$ for 200 mg N, $KH_2PO_4$ for 125 mg K and 100 mg P, Fe-EDTA for 2.5 mg Fe, and $ZnSO_4.7H_2O$ for 5 mg Zn kg$^{-1}$ of soil. For each kg of soil, 2,000 mycorrhizal spores (M1, M2, and M3 mixed AMF cultures) were applied to the soil 3 cm below the seedbed, and the maize seeds were sown at a depth of 5 cm. The AMF culture, designated M4, was applied to the seeds at a rate of 2.5 mg per 1 g of seed. Then the soil was covered, eight seeds were planted in the pots, and the planting process was completed by adding more soil. The number of plants per pot decreased to four when the plants emerged. Throughout the experiment, the plants were kept at 25/17 °C, 800 μmol m$^{-2}$ s$^{-1}$, 65% humidity, and a light/dark cycle of 14/10 h. The plants were irrigated with deionized water at field capacity until harvest, and 200 mg N kg$^{-1}$ of soil was added as top fertilizer at the 5–6 leaf stage.

## Low- and high-temperature applications

Temperature treatments were initiated when the plants were six weeks old (BBCH stage 16–18). Optimum mycorrhizal colonization is generally achieved when the maize plants are 6 weeks old. The determined low and high temperatures represent the temperature range that the maize plant is likely to encounter in the field at this stage.

Plants at 25 °C without mycorrhiza (M0) and with mycorrhiza (M1, M2, M3, and M4) were harvested directly for analysis after six weeks. For low- and high-temperature applications, plants were placed in a plant growth cabinet with a temperature adjustable

between 0 and +60 °C and kept at 5 °C, 10 °C, 15 °C, 35 °C, 40 °C, and 45 °C for 24 h (14 h of light; 10 h of darkness). After 24 h, the plants at each temperature were taken out of the chamber and harvested for analyses.

## Determination of mycorrhizal colonization in roots

Following the plants were harvested, the roots were meticulously separated from the soil and thoroughly washed with tap water to eliminate any adhering soil particles. Subsequently, 0.5–1 g pieces of the cleaned roots were meticulously excised and placed in AFA (alcohol (ethanol), formaldehyde, acetic acid) fixation liquid (90 mL 70% alcohol + 5 mL formalin + 5 mL acetic acid). The roots were then stored in the liquid at +4 °C until staining. To ascertain the presence of mycorrhizal fungi and the percentage of colonization, roots preserved in AFA fluid were stained with trypan blue. The staining solution was composed of lactic acid (40 mL), glycerin (80 mL), and purified water (40 mL), with the addition of 0.05% trypan blue. During the staining procedure, the roots were immersed in 10% KOH for a duration of half a day, followed by 10% HCl for a period of half an hour. Thereafter, they were transferred into the dye solution and left there for 2 h. Subsequently, the roots were heated in hot water at 50 °C for 5 min and then washed with pure water. Following this, the roots were transferred from pure water and immersed in lactic acid for 1 h. After that, they were removed from the lactic acid and made ready for microscopic inspection. The procedural steps described herein are modified from *Phillips & Hayman (1970)*.

The grid-line intersect method was used to determine the percentage of AMF colonization in roots stained with trypan blue (*Giovannetti & Mosse, 1980*). Approximately 0.5 g of the stained capillary roots were sampled and cut into 1–1.5 cm pieces, which were evenly distributed in 1 cm$^2$ areas in a plastic Petri dish and examined under a stereomicroscope. During the stereoscopic examination, one button was pressed for each root segment perpendicular to the intersection grids in the Petri dish, and two buttons were pressed together if AMF propagules (hyphae, vesicles, chlamydospores) were present in that vertical root segment. The last digit on the first button multiplied by 2 gave the length of the 0.5 g root, and the following equation was used according to the percentage of fungal colonization.

$$\% \text{ AMF Colonization} = (\text{Number of roots colonized with AMF/Total Number of Roots}) \times 100.$$

## Determination of mineral element concentrations

The above-ground parts of the harvested plants were washed with tap water, then once with distilled water, followed by a 0.2 N HCl solution. This process was repeated twice with distilled water. The excess water was then removed *via* filter paper, and the plants were dried in an oven at 65 °C until a constant weight was reached. After the determination of the weights of the dried samples, they were ground in a tungsten-coated grinding mill. A quantity of 0.3 g of dried and ground samples was weighed and dissolved with $HNO_3$ in a microwave oven (CEM Mars6). The samples were transferred to a 25 mL volumetric flask,

cooled, and brought to the desired level with ultrapure water. The filtrates were then filtered with blue band filter paper and transferred to 25 mL polyethylene bottles. The total N in the filtrate was determined according to the Kjeldahl method, while the P was determined using the vanadomolybdate yellow color method with a spectrophotometer (Thermo AQA2000E). K, Ca, and Mg were determined *via* a flame photometer (BWB/XP2011), and Zn, Fe, Cu, and Mn were determined *via* an atomic absorption spectrophotometer (Analytik Jena, NovAA 350).

## Determination of malondialdehyde content, proline content, and antioxidant enzyme activities

The amount of malondialdehyde (MDA) in maize leaf tissues was determined through the method described by *Hodges et al. (1999)*. After extracting 0.1 g of fresh leaf sample with 2 mL of 1% trichloroacetic acid (TCA), the resulting supernatant, which was obtained by centrifugation at 12,000 rpm for 15 min at 25 °C, was treated with 0.5% thiobarbituric acid (TBA) and 20% TCA. The mixture was then incubated in a water bath at 95 °C for 30 min, the reaction was stopped by placing it in an ice bath, and the absorbance was measured at 532 and 600 nm wavelengths using a spectrophotometer. The MDA content was determined as $nmolgFW^{-1}$.

Proline analysis was performed as described by *Bates, Waldren & Teare (1973)*. The maize leaves were homogenized with 3% sulfosalicylic acid and centrifuged at 6,000 rpm for 10 min. The resulting supernatant was transferred to test tubes, and glacial acetic acid and acid ninhydrin were added. The samples were then incubated in a 100 °C water bath for 1 h. After cooling, toluene was added, and the samples were taken from the upper phase. The concentrations were determined at 520 nm *via* a spectrophotometer (Thermo-Aquamate) on a calibration curve prepared using pre-prepared proline standards. The proline content was calculated as μmol proline/g fresh weight.

Superoxide dismutase (SOD), glutathione reductase (GR), ascorbate peroxidase (APX), and catalase (CAT) antioxidant enzyme activities were measured using the *Cakmak & Marschner (1992)* technique. Two mL of K-P buffer (pH 7.6) was used to homogenize 0.2 g of plant leaves to extract the enzymes. After that, the homogenate underwent 20 min of centrifugation at 15,000 g at 4 °C. All measurements of enzyme activity were made using this supernatant. First, the protein content was ascertained using the *Bradford (1976)* method because enzyme activities were computed based on the protein content.

The activity of SOD was measured by a spectrophotometer at 560 nm after incubating a mixture of 0.1 mL of sample, 2.9 mL of K-P buffer (pH 7.6), 0.5 mL of sodium bicarbonate, 0.5 mL of methionine, 0.5 mL of riboflavin, and 0.5 mL of nitro blue tetrazolium chloride (NBT) in glass tubes in the light for 10 to 15 min or until a blue color developed. The unit of measurement was μmol/mg protein.

The GR activity was measured using a mixture of 700 μL of K-P buffer (pH 7.6), 100 μL of GSSG, 100 μL of sample, and 100 μL of NADPH. The absorbance was recorded at 340 nm for 1 min. The reaction rate was determined as $nmol\ mg\ protein^{-1}\ min^{-1}$ by removing non-enzymatic oxidation using the NADPH extinction coefficient $(6.2\ mM\ cm^{-1})$.
The APX activity was measured by adding 700 μL of K-P buffer (pH 7.6), 100 μL of $H_2O_2$, 100 μL of sample, 100 μL of ascorbic acid, and using a spectrophotometer for 1 min at 290 nm. The unit of measurement was μmol mg protein$^{-1}$ min$^{-1}$.

The CAT activity was determined by measuring the light absorption at 240 nm for 1 min following the addition of 800 μL of K-P buffer (pH 7.6), 100 μL of sample extract, and 100 μL of $H_2O_2$. The amount of enzyme was determined as nmol mg protein$^{-1}$ min$^{-1}$.

## Gene expressions of HSP70 and HSP90

Gene expression analyses from all the treatment groups were carried out by the following experimental steps. Total RNA from 100 mg of ground-up leaves frozen in liquid nitrogen was isolated using the RNeasy Plant Mini Kit (Qiagen, Hilden, Germany). RNA purity was measured using a nanodrop device (ND2000, Thermo Fisher Scientific, Waltham, MA, USA). cDNA was synthesized from RNA samples with an A260/A280 ratio of 2.0 or greater. If the ratio was less than the target value, the analysis of the RNA samples was repeated. After adjusting the observed RNA concentrations to 1,000 ng/μL, RT-PCR was used to synthesize cDNA. RQ1 RNase-free DNase (Promega, Madison, WI, USA) was used to remove any DNA contamination. Procomcure Biotech's (Thalgau, Austria) VitaScript™ First-strand cDNA Synthesis Kit (PCCSKU1301) was used to synthesize cDNA. The cDNA samples were then adjusted to 100 ng/μL, and RT-qPCR was used to measure the expression levels of the relevant gene products. The RT-qPCR conditions were as follows: 95 °C for 5 min, 40 cycles of 95 °C for 15 s, 60 °C for 30 s, and 72 °C for 30 s. The 2X Magic SYBR Kit (Procomcure, Salzburg, Austria) and the CFX Connect Real-Time PCR Detection System (Bio-Rad, Hercules, CA, USA) were used. The relative expression was calculated using the $2^{-\Delta\Delta Ct}$ approach (*Livak & Schmittgen, 2001*). Two technical and three biological replicates were used for each analysis. The specific primers for the HSP70 (Accession no: DY307167), HSP90 (Accession no: EE257979), and housekeeping (Accession no: J01238) genes for maize were designed using the Primer 3 program. Their sequence information can be found in *Eskikoy & Kutlu (2024)*.

## Statistical evaluations

The IBM SPSS (IBM Corp., Armonk, NY, USA) 26 software was utilized to execute all the statistical procedures and generate the figures. The study's data were subject to analysis of variance using a completely random factorial design. Graphs illustrating dry weight and mineral element concentrations are presented as line graphs, whereas all other parameters are shown as bar graphs with error bars representing the mean ± standard error. Pearson correlation analysis was performed to understand the relationships between the variation in HSP70 and HSP90 gene expression and mycorrhizal colonization, antioxidants, and mineral element concentrations.

# RESULTS

## Mycorrhiza colonization rate

The control plants (M0) grown in sterilized soil showed no AMF colonization. The colonization rates obtained from different mixed AMF cultures of maize plants ranged
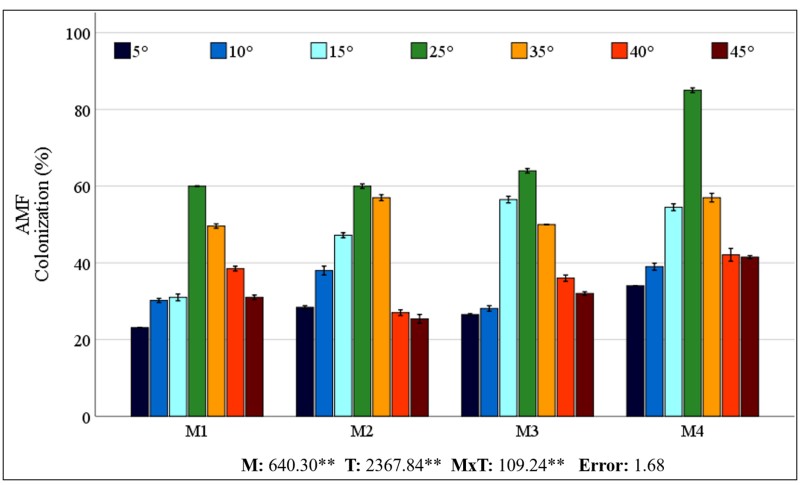

**Figure 1 Changes in mycorrhizal colonization of maize roots with temperature.** Mean squares obtained from ANOVA for the effects of mycorrhiza (M), temperature (T), and their interaction (M × T) on the measured variable. **$P < 0.01$.

from 60% to 85% at 25 °C (Fig. 1). However, low and high temperatures caused a decrease in colonization rates. The colonization rate declined in low-temperature environments relative to that in high-temperature conditions. A decrease in temperature to 15 °C resulted in the most rapid colonization rate of M3. At temperatures of 5 °C and 10 °C, M4 showed greater colonization. In comparison with the 25 °C, the decrease in colonization at low temperatures was comparatively less pronounced in M2. At high temperatures, M4 had the highest colonization rate, whereas the decrease in colonization with increasing temperature was less pronounced in M1.

## Dry weight and mineral element concentrations

The statistical significance ($P < 0.01$) of the interactions among the mycorrhiza, temperature, and mycorrhiza × temperature treatments was significant for the dry weight of the maize plants (Fig. 2A). The highest value was obtained from the maize plants in the M3 mixed AMF culture, while the lowest value was determined in the group where mycorrhiza was not applied (M0), considering the mean of the mycorrhiza treatments. When the mean dry weights of the maize plants in the various temperature treatments were analyzed, the highest value (20.76 g plant$^{-1}$) was obtained at 25 °C, whereas the lowest value (13.88 g plant$^{-1}$) was obtained at 5 °C (Fig. 2A). For the mycorrhiza × temperature interaction, the highest value (23.99 g plant$^{-1}$) was determined at 25 °C with M3 mixed AMF culture, and the lowest value (8.30 g plant$^{-1}$) was determined at 45 °C in the group without mycorrhiza (M0).

The statistical analysis of the macro mineral nutrient element concentrations revealed a statistically significant interaction effect on maize plants treated with combinations of mycorrhiza, temperature, and mycorrhiza × temperature ($P < 0.05$ and $P < 0.01$). The mineral element concentrations in the maize plants varied according to the mixed AMF cultures. The M4 mixed AMF culture had the highest concentrations of N, P, and K, while

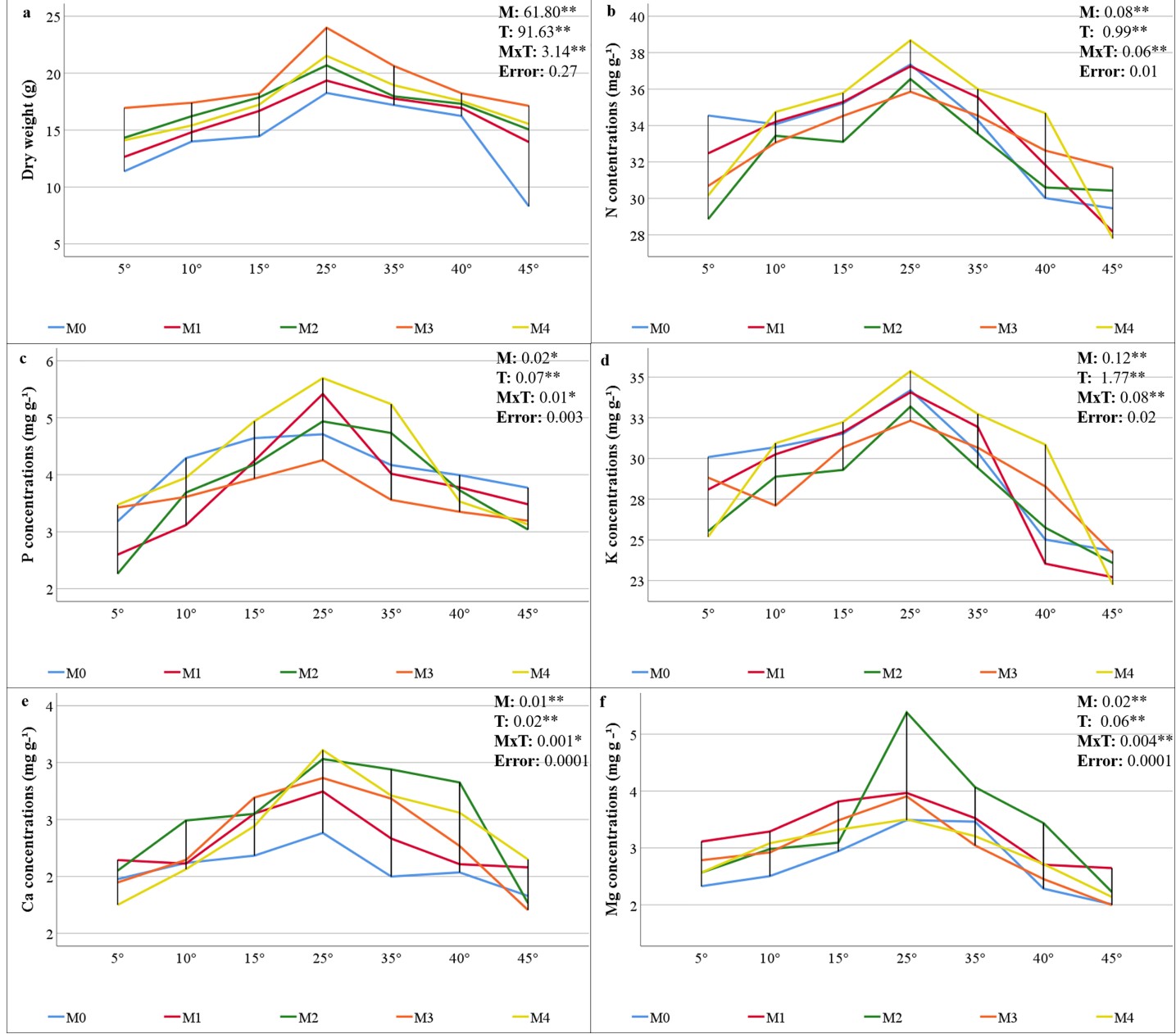

**Figure 2 Changes in dry weight (A) and macro element (N (B), P (C) K (D), Ca (E), and (Mg (F)) concentrations in maize plants with mycorrhiza and temperature treatments.** Mean squares obtained from ANOVA for the effects of mycorrhiza (M), temperature (T), and their interaction (M × T) on the measured variable. $^*P < 0.05$, $^{**}P < 0.01$.

the M2 mixed AMF culture had the highest concentrations of Ca and Mg (Figs. 2B–2F). The highest concentrations of N, P, K, Ca, and Mg were detected in maize plants cultivated at 25 °C. Nutrient concentrations decreased when the temperature was above or below 25 °C. At temperatures of 15 °C and 35 °C, the mycorrhiza treatments resulted in nutrient concentrations that approximated the 25 °C level observed in the maize plants. According to the mycorrhiza × temperature interaction, the N, P, K, and Ca concentrations were highest at M4 × 25 °C, and the Mg concentration was highest at M2 × 25 °C.

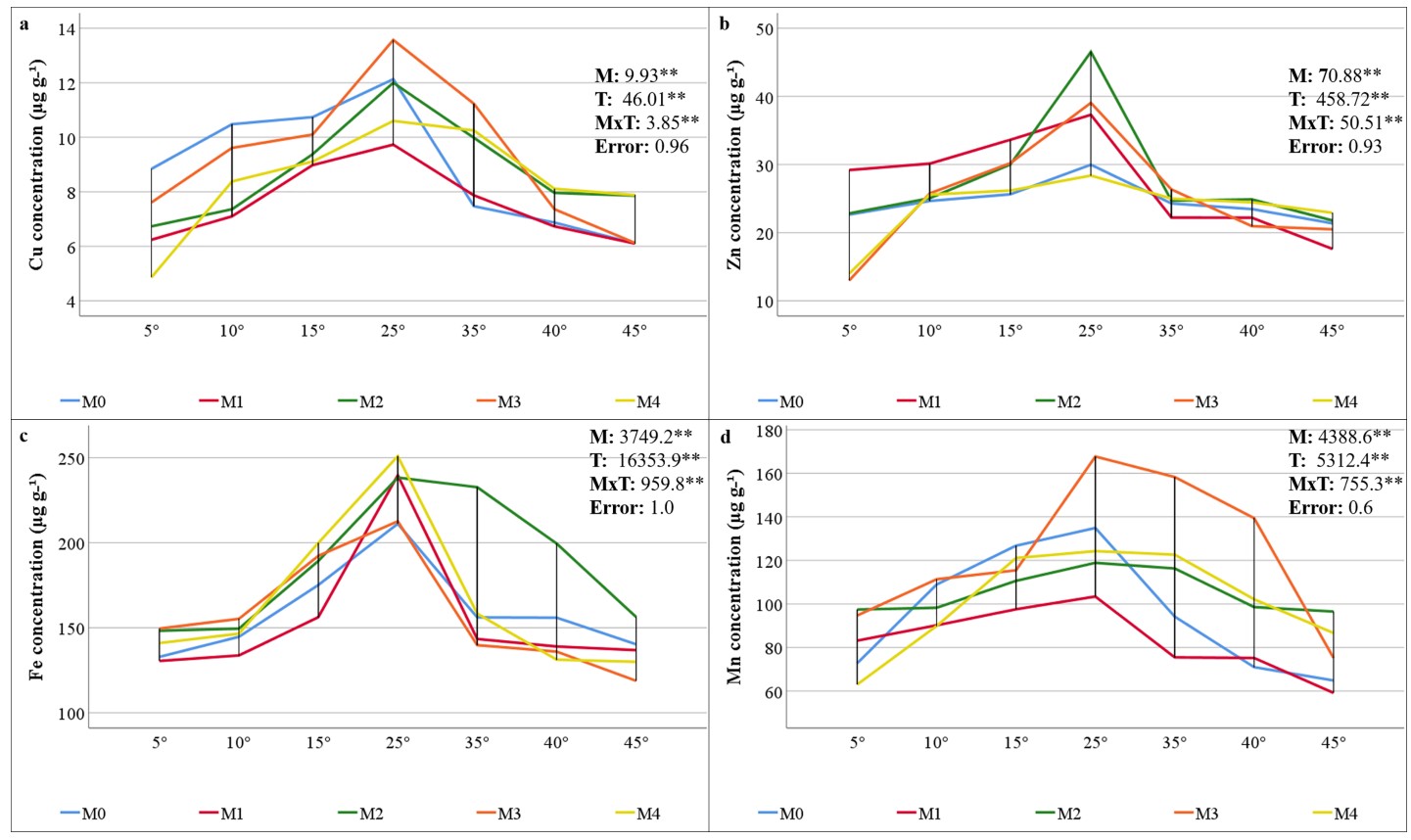

**Figure 3 Changes in micro element (Cu (A), Zn (B), Fe (C), and Mn (D)) concentrations in maize plants with mycorrhiza and temperature treatments.** Mean squares obtained from ANOVA for the effects of mycorrhiza (M), temperature (T), and their interaction (M × T) on the measured variable. **$P < 0.01$.          

The results of the variance analysis of the micronutrient element (Cu, Zn, Fe, and Mn) concentrations of maize plants inoculated with mycorrhiza under different temperature treatments indicated that mycorrhiza, temperature, and the interaction of mycorrhiza × temperature were statistically significant ($P < 0.01$). The mineral element concentrations present in the maize plants varied depending on the mixed AMF culture. The M3 mixed AMF culture had the highest concentrations of Cu and Mn, whereas the M2 mixed AMF culture had the highest concentrations of Fe and Zn (Figs. 3A–3D). In terms of temperature applications, the maize plants presented the highest concentrations of Cu, Fe, Mn, and Zn at 25 °C. It was determined that a decrease occurred when the temperature exceeded or fell below 25 °C. According to the mycorrhiza × temperature interaction, the highest concentrations of nutrients were found for Zn at M2 × 25 °C, Cu and Mn at M3 × 25 °C, and Fe at M4 × 25 °C.

## Malondialdehyde content, proline content, and antioxidant enzyme activities

The contents of MDA, which is the end product of lipid peroxidation, are shown in Fig. 4A. The findings reveal a statistically significant relationship between mycorrhiza,

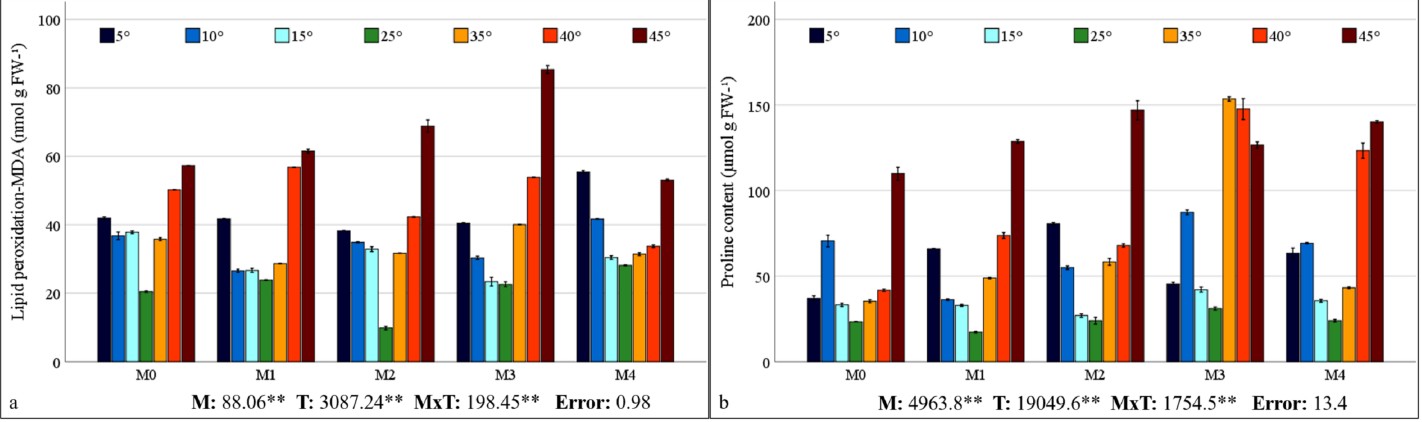

**Figure 4 Changes in lipid peroxidation (A) and proline content (B) in maize plants with mycorrhiza and temperature treatments.** Mean squares obtained from ANOVA for the effects of mycorrhiza (M), temperature (T), and their interaction (M × T) on the measured variable. **$P < 0.01$.

temperature treatments, and their interaction on the MDA content of maize plants. As the severity of cold or heat stress increased, the MDA contents of the maize plant also increased. The results indicated that the MDA value was lowest in all groups under 25 °C and increased with temperature changes. The highest MDA value (85.36 nmol g$^{-1}$ FW) was ascertained at 45 °C with M3 mixed AMF culture, while the lowest value (9.82 nmol g$^{-1}$ FW) was determined at 25 °C with M2 mixed AMF culture.

The alteration in proline content of the maize plant in response to mycorrhiza and temperature treatments is statistically significant ($P < 0.01$), and there is a significant interaction between the treatments (Fig. 4B). The proline content in the maize plant peaked at 35 °C with M3 mixed AMF culture, reaching 153.33 nmol g$^{-1}$ FW, and was the lowest (17.17 nmol g$^{-1}$ FW) at 25 °C with M1 mixed AMF culture. Proline content was low at 25 °C, and it increased in response to temperature changes. Proline content has also increased with mycorrhiza applications at 25 °C, except for the M1 mixed AMF culture. Proline accumulation is more pronounced at high temperatures than at low temperatures.

Mycorrhiza, temperature, and the interaction between mycorrhiza and temperature were identified as having a statistically significant effects ($P < 0.01$) on the antioxidant enzyme activities of the maize plants (Figs. 5A–5D). The highest values for SOD, GR, and APX were obtained from maize plants treated with the M1 mixed AMF culture, while the highest values for CAT activity were obtained from the M3 culture. With respect to the total change in activity depending on temperature, SOD activity was observed to be the highest at 5 °C, GR activity at 35 °C, APX activity at 10 °C, and CAT activity at 40 °C. The SOD activity of the maize plants was the highest (537.75 μmol protein mg$^{-1}$) at 35 °C, where the M1 mixed AMF culture was applied, and the lowest value (4.75 μmol protein mg$^{-1}$) was determined at 25 °C with M2 application (Fig. 5A). It was determined that SOD activity was low in maize plants at 25 °C and increased with temperature changes. The highest GR activity (318.27 nmol mg protein$^{-1}$ min$^{-1}$) was detected at 45 °C with M1 application, and the lowest (19.09 nmol mg protein$^{-1}$ min$^{-1}$) was detected at

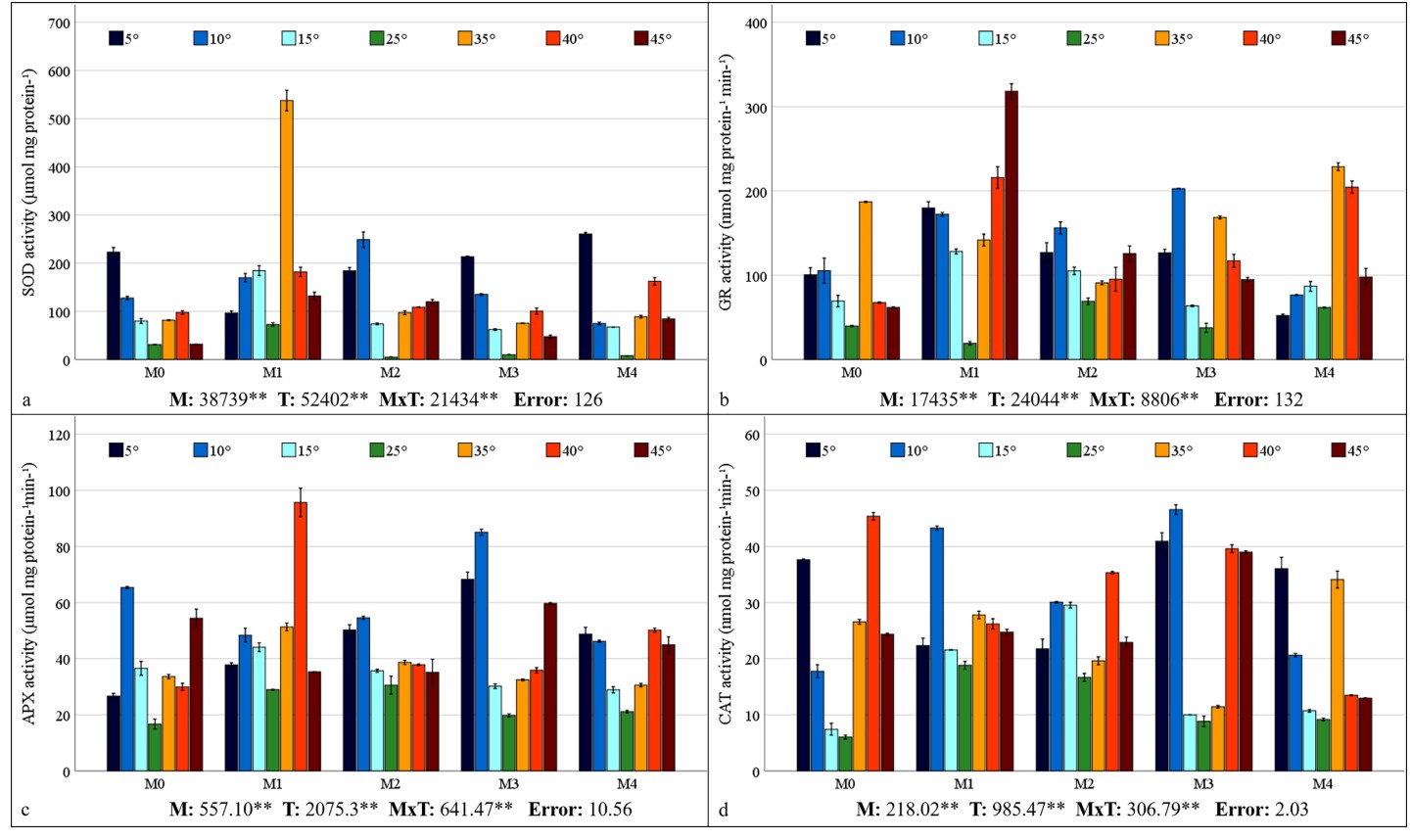

**Figure 5** **Changes in antioxidant enzyme (SOD (A), GR (B), APX (C), and CAT (D)) activity in maize plants with mycorrhiza and temperature treatments.** Mean squares obtained from ANOVA for the effects of mycorrhiza (M), temperature (T), and their interaction (M × T) on the measured variable. **$P < 0.01$.

25 °C with M1 application (Fig. 5B). It was determined that the GR activity was the lowest at 25 °C and that GR activity increased based on temperature changes. The maximum APX activity of the maize plants (95.70 nmol mg protein$^{-1}$ min$^{-1}$) was observed at 40 °C with the M1 mixed AMF culture, and the minimum (16.67 nmol mg proteinn$^{-1}$ min$^{-1}$) APX activity was recorded at 25 °C in M0 (Fig. 5C). APX activity increased with mycorrhiza applications and temperature changes. The highest CAT activity (46.57 nmol mg$^{-1}$ protein mg$^{-1}$ min) was determined at 40 °C with M3 application, and the lowest (6.02 nmol mg$^{-1}$ protein mg$^{-1}$ min) was determined at 25 °C in M0 (Fig. 5D). The study revealed that CAT activity reached its lowest point at 25 °C and increased in response to temperature variations. Furthermore, the application of mycorrhizae led to an augmentation in CAT activity at 25 °C.

## HSP70 and HSP90 gene expressions

HSP70 and HSP90 gene expressions in maize plants were found to be statistically significant at the 1% level under mixed AMF culture, temperature treatment, and the interaction of these two factors (Fig. 6). The highest levels of HSP70 and HSP90 expression in maize plants were observed in the M2 mixed AMF culture. The highest

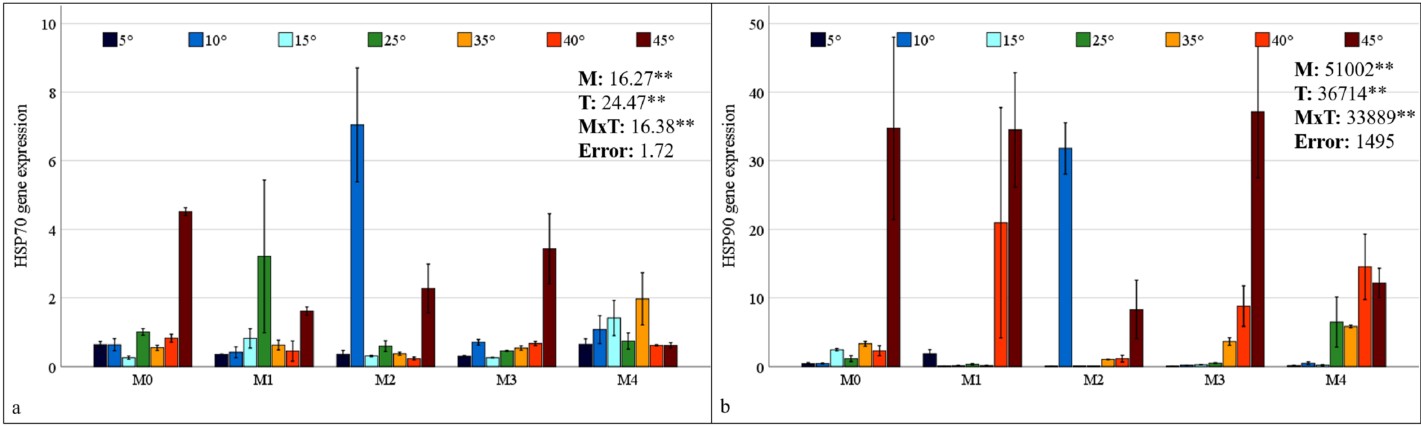

**Figure 6 Changes in HSP70 (A) and HSP90 (B) gene expression in maize plants with mycorrhiza and temperature treatments.** Mean squares obtained from ANOVA for the effects of mycorrhiza (M), temperature (T), and their interaction (M × T) on the measured variable. **$P < 0.01$.

expression values were recorded at low temperatures of 10 °C and high temperatures of 45 °C (Fig. 6).

In the M0 group, the expression of the HSP70 gene in maize plants decreased at low temperatures and at 35 °C and 40 °C; however, it increased approximately 2.5 times at 45 °C compared with that at 25 °C. In the M1 group, mycorrhiza application resulted in an increase in HSP70 gene expression compared to the M0 group at 25 °C. At both low and high temperatures, the application of the M1 mixed AMF culture resulted in a decrease in the gene expression of the maize plants relative to 25 °C (Fig. 6A). In the M2 group, HSP70 gene expression in maize plants at low temperatures was downregulated at 15 °C and 5 °C compared with that at 25 °C, whereas it increased approximately 7-fold at 10 °C. At high temperatures, the HSP70 gene expression level in maize plants decreased slightly at 35 °C and 40 °C, but increased at 45 °C. In the M3 group, the highest HSP70 gene expression in maize plants was observed at 45 °C. The alterations at other temperature levels were negligible. In the M4 group, the HSP70 gene expression of the maize plants increased at 15 °C compared to 25 °C but decreased in parallel with the decrease in temperature. Similarly, at high temperatures, the expression level, which exhibited an increase at 35 °C, demonstrated a decrease with rising temperature (Fig. 6A).

In the M0 group, the expression of the HSP90 gene in maize plants increased at 15 °C compared to 25 °C; however, it decreased to very low levels at 5 °C and 10 °C. The expression level increased at high temperatures. Notably, at 45 °C, the expression level demonstrated a marked increase, reaching approximately 35-fold that of 25 °C. In the M1 group, the significant upregulation of the HSP90 gene in maize plants at 40 °C and 45 °C was a notable finding (Fig. 6B). In the M2 group, HSP90 gene expression in maize plants at low temperatures, approximately 25 °C at 15 °C and 5 °C; however, it increased approximately 35-fold at 10 °C. At high temperatures, HSP90 gene expression levels in maize plants increased beginning at 35 °C and 40 °C, reaching an approximately 9-fold increase at 45 °C. In the M3 group, the highest HSP90 gene expression in the maize plants

**Table 1** Correlations of examined characters with gene expressions of HSP70 and HSP90.

|  | HSP70 | HSP90 |
|---|---|---|
| AMF colonization | −0.30 | −0.20 |
| Plant dry weight | −0.21 | −0.30 |
| N concentration | −0.07 | −0.41* |
| P concentration | 0.04 | −0.22 |
| K concentration | −0.14 | −0.52** |
| Ca concentration | −0.13 | −0.30 |
| Mg concentration | −0.21 | −0.46** |
| Fe concentration | −0.10 | −0.38* |
| Zn concentration | −0.07 | −0.34* |
| Mn concentration | −0.21 | −0.39* |
| Cu concentration | −0.23 | −0.45** |
| MDA content | 0.26 | 0.62** |
| Proline content | 0.14 | 0.52** |
| SOD activity | 0.004 | −0.05 |
| GR activity | −0.01 | 0.34* |
| APX activity | 0.09 | 0.30 |
| CAT activity | 0.10 | 0.10 |

Notes:
* $P < 0.05$.
** $P < 0.01$.

was observed at 45 °C. Conversely, increases at other high temperatures were observed at a lower magnitude. Under low-temperature stress, the expression level of the HSP90 gene approaches an undetectable level. In the M4 group, the HSP90 gene expression in the maize plants diminished to an undetectable level at low temperatures. At high temperatures, the expression level, which was close to 25 °C at 35 °C, demonstrated an increase at 40 °C and a subsequent decrease at 45 °C (Fig. 6B).

### Correlations between the examined parameters and gene expressions

The correlations between mycorrhizal colonization, plant dry weight, nutrient element concentrations, and antioxidant enzyme activities examined in the study, and HSP70 and HSP90 gene expressions are presented in Table 1. While the examined characteristics did not demonstrate a significant correlation with HSP70 gene expression, HSP90 gene expression revealed negative and significant correlations with N, K, Mg, Fe, Zn, Mn, and Cu concentrations, as well as positive and significant correlations with MDA, proline content, and GR activity.

## DISCUSSION

### Mycorrhiza colonization rate

In the root region of the maize plant, the greatest degree of colonization in terms of root infection was detected in the M4 mixed AMF culture at 25 °C (85%). Mycorrhizal colonization was negatively impacted by decreases in temperature, while temperature

increases resulted in a corresponding decrease in mycorrhizal colonization (Fig. 1). *Zhu, Song & Liu (2016)* reported that mycorrhizal (*Glomus tortuosum*) colonization decreased from 49% to 42% when the temperature dropped from 25 °C to 15 °C. Low temperatures hinder AMF growth, affecting the rate of mycorrhizal colonization of seedlings, which then reduces the AMF colonization rate. Earlier research showing reduced mycorrhizal colonization of plants under low temperatures aligns with the findings of this study (*Liu et al., 2016b*; *Zhu, Song & Liu, 2017*; *Hu et al., 2024*). This inhibition may result from low temperatures slowing AMF mycelium and root growth, decreasing interactions, and thus mycorrhizal colonization (*Hu et al., 2024*).

A general increase in temperature up to 30 °C generally promotes AMF development (*Gavito et al., 2005*). The impact of high temperatures on mycorrhizal colonization has been observed to be either adverse or inconsequential, as evidenced by *Zhu et al. (2011)*, *Mathur et al. (2021)*, and *Liu et al. (2023)*. The colonization percentage of M4 exhibited a higher frequency than the other types across all temperature levels. The higher colonization rate of M4 may be due to its greater number of species. Because each species has different temperature tolerances, even if some species are lost due to stress, others may continue their symbiosis with the plant. This finding also demonstrates the coping strategy of M4-inoculated maize plants, which allocate more carbon to AMF under stress conditions, thereby increasing the AMF biomass in the soil. In a previous study, *Hussain et al. (2021)* reported that coating seeds with AMF increased root colonization in maize plants. This outcome may have been achieved by the application of M4 to the seeds, in contrast to other mixed AMF cultures applied to the soil. While the cited studies have generally focused on a single AMF species and used different species, mycorrhizal colonization has been reported to decrease as a result of cold or heat stress. Different species may have different temperature tolerance thresholds, so using a mixture of species would be logical. A mixture of AMF species is expected to be more effective in occupying ecological niches and adapting to environmental changes (*Amir & Crossay, 2024*). Synergistic effects of AMFs used in mixed culture should also be considered. Although M4 culture had the highest colonization rate under all temperature conditions, the decrease in colonization was also high under stress conditions. Considering the extent of the decrease in colonization indicates that the M2 mixed AMF culture demonstrated a higher degree of tolerance to low temperatures, while the M1 mixed AMF culture exhibited a higher degree of tolerance to high temperatures. It has been reported that the species *Glomus intraradices*, *Glomus constrictum*, and *Glomus microcarpum*, which constitute M2, have a high ability to form colonies together (*Burak et al., 2024*). Despite the absence of prior data concerning the temperature responses of these species, the synergistic effect they create and their sufficient symbiotic relationship with the maize plant have enabled them to better maintain their colonization rates under cold stress. *Glomus intraradices*, which forms M1, is the most widespread model mycorrhiza among *Glomus* species. The *Glomus mosseae* species has been reported to be more common in warm and temperate regions than in cold or dry regions (*Khazna, Bibi & Abu-Dieyeh, 2024*). It is hypothesised that, owing to these characteristics, colonization rates may decrease less at high temperatures. Interpretations of changes in root colonization rates, which can be considered an indicator

of the tolerance of mixed AMF cultures to low and high temperatures, may also affect changes in the maize plant under stress conditions.

## Dry weight and mineral element concentrations

The findings indicated that the optimal temperature conditions (25 °C) resulted in the greatest increase in plant dry weight, while temperature fluctuations led to a decline in plant dry weight. Mycorrhiza application at 25 °C increased the dry weight of mycorrhiza-treated maize plants compared to the M0 group. This outcome aligns with earlier research that also revealed increased dry weight in the aboveground parts of plants following mycorrhiza application (*Hussain et al., 2021*). Furthermore, the application of different mycorrhiza species applications has been shown to result in higher dry matter production compared to the control (*Ma et al., 2022*). Mycorrhiza-inoculated maize plants exhibited greater plant dry weight than non-mycorrhizal plants under low and high temperatures in all groups. The accumulation of dry matter exhibited a reciprocal relationship with temperature, decreasing in proportion to the increase and decrease in temperature from 25 °C. However, these decreases were less pronounced in plants inoculated with mycorrhiza. The most effective inoculum was found to be the M3 mixed AMF culture, which exhibited of the highest dry weight at all temperatures (Fig. 2A). This phenomenon can be attributed to the enhanced nutritional quality of maize, which is attributable to the application of different mixed AMF cultures. Earlier studies indicate that the dry matter rate increases due to the root area and length of the plant, water uptake, and water storage and transportation, which are facilitated by the production of other substances in the plant with which mycorrhiza (especially under heat stress *Glomus fasciculatum*) are in a symbiotic relationship (*Wahab et al., 2023*). It has been documented that plant growth is reduced under conditions of heat stress; however, there is a significant increase in shoot dry weight in plants inoculated with mycorrhiza (*Mathur et al., 2021*). Moreover, *Zhu, Song & Xu (2010)* reported that shoot dry weight was higher under low and high temperature stress in plants inoculated with mycorrhizae (*Glomus etunicatum*) compared to non-inoculated plants. In contrast to the high dry shoot weight of M3 under all temperature conditions, the least dry matter loss was observed in M2 at 5 °C and in M1 at 45 °C, consistent with root colonization.

The hyphae formed by AMF in their mutually beneficial relationship with the roots of maize plants create a vital link between plants and soil that is essential for nutrient exchange. While plants provide the essential carbon sources required by AMF, in return, AMF significantly enhance the plants' ability to absorb nutrients from the soil, which is particularly vital under nutrient-deprived conditions caused by low or high temperature stress (*Nie et al., 2024*). The study determined the N concentration of maize was within the range of 27.81–38.68 mg g$^{-1}$ (Fig. 2B). The critical levels for N content in maize plants were found to be in the range of 35–50 mg g$^{-1}$ (*Jones, Wolf & Mills, 1991*), and overall, it was determined that the N concentration in maize was sufficient in the temperature range of 15 °C to 35 °C. It has been observed that as temperatures increase or decrease, there is a decrease in N uptake, which leads to a corresponding decrease in N concentration. However, the application of mycorrhizal inoculation has been shown to enhance N uptake,
with a substantial portion of N being transferred to the host plant under low or high temperatures (*Liu et al., 2016a*; *Loo et al., 2022*; *Thangavel et al., 2022*). Although the N concentration fell below the critical threshold in all applications at the extreme temperatures of 5 °C and 45 °C, it remained within acceptable limits for the M4 application at the 10 °C to 40 °C range.

The critical levels of P content for maize plants range from 3 to 5 mg g$^{-1}$ (*Jones, Wolf & Mills, 1991*). The study revealed that maize P concentrations ranged from 2.26 to 5.69 mg g$^{-1}$, indicating sufficient levels (Fig. 2C). While temperatures of both low and high ranges resulted in lower P concentrations than did 25 °C, P levels were elevated in plants that had been inoculated with mycorrhizae. The AMF enzymatically breaks down organic P in the soil and transfers inorganic P to the host plant *via* the mycelial network. This activity has been demonstrated to enhance the uptake of P under conditions of stress, thereby promoting improved crop yield (*Xu et al., 2018*; *Jajoo & Mathur, 2021*; *Ahmed et al., 2025*).

The findings indicate that the decline in N and P concentrations at low and high temperatures, along with their enhancement through AMF inoculation, applies to other macronutrients. *Jones, Wolf & Mills (1991)* reported that the critical concentrations of K, Ca, and Mg in maize plants are 25–40, 3–7, and 1.5–4.5 mg g$^{-1}$, respectively. The study revealed that the K concentrations ranged from 22.26 to 35.37 (Fig. 2D), and the Mg concentrations ranged from 2.0 to 5.4 mg g$^{-1}$ (Fig. 2E), both of which were found to be within acceptable limits. The concentrations of Ca were found to be inadequate, ranging from 1.7 to 3.1 mg g$^{-1}$ (Fig. 2F). The application of mycorrhizae at low and high temperatures enhanced the uptake of the aforementioned macronutrients. These results align with those previously reported in related studies (*Al-Amri, 2021*; *Khan, Shah & Tian, 2022*; *Thangavel et al., 2022*). *Glomus mosseae, Glomus etunicatum*, and *G. margarita* are more effective at absorbing soluble nutrients. It has been documented that these species have a higher efficacy rating than others due to their superior capacity for water uptake (*Carrara & Heller, 2022*). The critical levels for Cu, Fe, and Mn concentrations in maize plants have been reported as 5–20, 50–250, and 20–300 µg g$^{-1}$, respectively (*Jones, Wolf & Mills, 1991*). The findings demonstrate that the levels of Cu ranged from 4.86 to 13.57 µg g$^{-1}$ (Fig. 3A), Fe ranged from 118.82 to 250.88 µg g$^{-1}$ (Fig. 3C), and Mn ranged from 59.17 to 167.70 µg g$^{-1}$ (Fig. 3D). This suggests that there were sufficient levels of these elements at all temperature levels. However, the concentrations of these micronutrients were found to decrease due to low and high temperature stress. M3 mixed AMF cultures resulted in an increase in the Cu concentration at 25 °C, while the other combinations caused a decrease. At low temperatures, mycorrhizal inoculation generally caused a decrease in the Cu concentration. However, when compared to non-mycorrhizal plants under high temperature conditions, an increase in the Cu concentration was observed in all the maize plants where mycorrhiza was applied at 35 °C. Furthermore, an increase in the Cu concentration was observed in maize plants where M2, M3, and M4 mixed AMF cultures were applied at 40 °C and 45 °C. In contrast to the findings of our study, *Liu et al. (2016a)* reported that plants treated with mycorrhiza under low-temperature stress exhibited higher concentrations of Cu compared to the plants that did not receive

mycorrhiza treatment. Similarly, *Zhu, Song & Liu (2017)* have found that mycorrhizal inoculation increases the Cu concentration at high temperatures. The concentrations of Fe and Mn decreased at both low and high temperatures. However, the presence of mixed AMF cultures resulted in fluctuations in the Fe and Mn concentrations, with the direction of change depending on the specific temperature conditions. Many studies have previously reported that the application of mycorrhiza does not affect the uptake of Fe and Mn (*Wu & Zou, 2010*; *Zhu, Song & Liu, 2017*).

In the present study, Zn concentrations ranged from 12.98 to 46.53 mg kg$^{-1}$. Given that the Zn content in maize plants within the range of 20–60 mg kg$^{-1}$ (*Jones, Wolf & Mills, 1991*) is critical, it was determined that the Zn concentration in maize plants inoculated with M3 and M4 mixed AMF cultures was below the sufficient level threshold (20 mg kg$^{-1}$) under 5 °C. At temperatures of 10 °C, 15 °C, and 25 °C, the presence of AMF resulted in elevated Zn concentrations. Conversely, at 5 °C, the application of M3 and M4 led to reduced Zn concentrations. At higher temperatures, the utilization of M1 and M3 resulted in decreased Zn concentrations (Fig. 3B). Higher Zn concentrations with mycorrhiza applications have been reported in previous studies (*Liu et al., 2016a*; *Loo et al., 2022*; *Ortaş, 2025*).

The composition of the mixed AMF culture, which is conducive to attaining elevated concentrations of each nutrient element, is temperature-dependent. The highest N, P, and K concentrations were found in the M4 group; the highest Ca, Mg, Zn, and Fe concentrations in the M2 group; and the highest Cu and Mn concentrations in the M3 group. This may be related to the fact that the mycorrhizal species contained in the mixed AMF cultures facilitate the uptake of these nutrients. Temperature stress reduced the concentrations of the nutrients. These alterations in nutrient concentrations may be attributable to decreased mycorrhizal colonization under conditions of stress and temperature-dependent changes in nutrient uptake and translocation in plants. At 25 °C, the highest concentrations of N, P, K, Ca, and Fe were obtained with the M4 mixed AMF culture. M2 and M3 resulted in the highest concentrations of other nutrient elements. At a temperature of 5 °C, the concentrations of N, K, and Cu were found to be highest in the absence of mycorrhiza. P concentrations exhibited elevated levels in the absence of mycorrhiza at temperatures of 10 °C, 40 °C, and 45 °C. To ascertain the most suitable recommendation in light of these results, which vary depending on the mixed AMF culture and temperature levels, decisions must be made based on concentrations that do not fall below critical levels.

## Malondialdehyde content, proline content, and antioxidant enzyme activities

Abiotic factors, such as temperature stress, can accelerate the production of ROS, which consequently leads to adverse oxidative conditions, including cell damage and membrane lipid peroxidation (*Sachdev et al., 2021*). ROS attacks nucleic acids, lipids, and proteins, and the degree of damage depends on the balance between the formation of ROS and its removal of this formation by antioxidants. To defend plants against oxidative damage, antioxidant enzymes such as SOD, APX, CAT, and GR prevent damage caused by ROS

(*Das & Roychoudhury, 2014*). Many studies have demonstrated that mycorrhiza facilitate plants in increasing water and nutrient uptake in plants under conditions of low and high temperature stress. This process serves to protect plants against oxidative damage and enhances their ability to prevent stress-related effects by increasing osmolyte accumulation (*Zhu, Song & Liu, 2017*).

In all the AMF applications, the MDA content increased as the temperature decreased and increased. Previous studies have determined that MDA values increase in maize plants at high and low temperatures (*Cao et al., 2024*; *Hu et al., 2024*; *Meng et al., 2025*). The impact of AMF application on the MDA content revealed a decline in the M2 group compared to the M0 group at 25 °C (Fig. 4A). A comparison of mixed AMF cultures at low temperatures revealed a decrease in the MDA content in maize plants treated with all mixed AMF cultures. This decrease was observed in comparison to the M0 group at 5 °C for M2 and M3, at 10 °C for M1, M2, and M3, and in all mixed AMF cultures at 15 °C. As demonstrated in the previous research, mycorrhizal inoculation such as *D. versiformis*, *G. etunicatum* and *G. intraradices* species has been shown to reduce MDA levels in leaves under low-temperature stress (*Chen et al., 2014*; *Hu et al., 2024*). A comparison of mixed AMF cultures at high temperatures revealed a decline in the MDA content in maize plants treated with M1, M2, and M4 at 35 °C; M2 and M4 at 40 °C; and M4 at 45 °C, compared with those in the M0 group. As indicated by *Mathur et al. (2021)*, mycorrhizal inoculation, including *Glomus* species, of maize plants subjected to high temperatures resulted in diminished MDA levels compared with those in non-inoculated plants. In another study, it was determined that MDA levels were lower in mycorrhizal (*G. etunicatum*) maize plants compared to non-mycorrhizal plants under both low and high temperature stress conditions (*Zhu, Song & Xu, 2010*). In general, MDA accumulation was lowest in maize plants inoculated with M3 at low temperatures and in those inoculated with M4 at high temperatures.

An increase in leaf proline content was observed at low and high temperatures in comparison with 25 °C (Fig. 4B). It has been reported that the increase in proline content observed in response to stress improves osmotic regulation and that this mechanism is preserved in mycorrhizal symbiosis (*Mitra, Djebaili & Pellegrini, 2021*). The alterations in proline content observed in plants within the M3 group at low temperatures closely resembled those in the M0 group. The decrease in the proline content observed at 5 °C compared with 10 °C indicates a reduction in the protective mechanism. The proline content of maize plants at 5 °C is higher than that of the group without mycorrhiza (M0). At 15 °C, the proline content of the plants in the M2 mixed AMF culture was lower. At 10 °C, a decrease in proline content was observed in maize plants treated with M1 and M2 mixed AMF culture compared to the group without mycorrhiza application (M0). As indicated by *Yan et al. (2021)*, the proline content has been observed to increase at low temperatures, particularly in conjunction with *Glomus intraradices* application. This increase is attributed to the accumulation of proline in the leaves, a process that serves to regulate osmotic pressure (*Chen et al., 2014*). A previous study revealed that the proline content in the leaves of mycorrhizal maize inoculated with *G. etunicatum* under low-temperature stress was lower than that in a group without mycorrhizal application

(*Zhu, Song & Xu, 2010*). Proline accumulation under high-temperature stress exhibited a parallel increase with the increase in temperature in the M1, M2, and M3 mixed AMF cultures. *Raza et al. (2023)* identified elevated levels of non-enzymatic antioxidants such as proline in plants under high temperatures. In addition, *Romero-Munar et al. (2023)* reported high proline accumulation in plants treated with mycorrhiza under high temperatures. The sudden increase in proline content at 35 °C in response to the M3 application suggests that this mixed AMF culture provides enhanced support for plant growth at these temperature values. Consequently, the sudden stress response in the plant, as evidenced by elevated proline accumulation, may have been precipitated by the beneficial effects of the symbiotic relationship. The highest accumulation of proline indicates that maize plants can protect themselves against stress caused by temperature changes in mycorrhizal symbiosis at extreme temperatures such as 5 °C and 45 °C was found in maize plants inoculated with M2.

The SOD activity decreased with M2, M3, and M4 applications compared to the M0 group at 25 °C (Fig. 5A). *Chen et al. (2014)* also determined that plants with mycorrhizal inoculation, especially *G. etunicatum* and *G. intraradices* at 25 °C had lower SOD activity compared to plants without mycorrhizal inoculation. SOD activity was higher in maize plants treated with M4 at 5 °C, M1, M2, and M3 at 10 °C, and M1 mixed AMF culture at 15 °C in comparison to the M0 group. *Yan et al. (2021)* ascertained that mycorrhizal (*Glomus intraradices*) plants evidenced elevated levels of SOD activity in comparison with uninoculated plants under conditions of low-temperature stress. In maize plants inoculated with mycorrhiza, SOD activity increased at temperatures of 35 °C, 40 °C, and 45 °C in comparison with the uninoculated (M0) group. As has been documented in prior reports, mycorrhizal inoculation has been shown to upregulate SOD activity under conditions of high temperature stress (*Jian et al., 2025*).

An increase in GR activity was determined in maize plants in which M2 and M4 mixed AMF cultures were inoculated at 25 °C compared to the M0 group (Fig. 5B). It was also reported by *Begum et al. (2022)* that mycorrhizal (*Glomus versiforme*) inoculation increased GR activity under non-stress conditions. The GR activity revealed a marked increase in plants treated with M1 at 15 °C and 5 °C and with M3 at 10 °C. A notable increase in GR activity, up to 10 °C, was observed in M0, M1, and M3, followed by a decline at 5 °C. The GR activity increased with decreasing temperature under the M1 application and decreased under the M4 application. *Wei et al. (2023)* reported that *Glomus intraradices* (*R. irregularis*) mycorrhiza applications improved GR activity due to low-temperature stress. Under high temperature conditions, the application of M4 at 35 °C and M1 at 40 °C and 45 °C resulted in the greatest GR activity. Conversely, the GR activity exhibited a decline with rising temperature in the M0, M3, and M4 treatments, while it demonstrated an increase in M1 and M2. The GR expression of M1 was elevated in both temperature conditions.

Mycorrhizal plants exhibited increased APX activity under low-temperature stress when compared to the non-mycorrhizal plants (Fig. 5C). This increase was observed at 5 °C in all the mycorrhizal plants and at 10 °C and 15 °C in the maize plants treated with M3 and M1 mixed AMF cultures, respectively. Previous studies have shown that

mycorrhizal (*Glomus intraradices* and *Glomus mosseae)* inoculation enhances APX activity under low-temperature stress (*Liu et al., 2017*; *Wei et al., 2023*). Mycorrhizae increased APX activity in maize plants at high temperatures. This was observed in plants with M1 and M2 at 35 °C, M0 at 40 °C, and M3 at 45 °C. *Haddidi et al. (2020)* reported similar findings in mycorrhiza (*Rhizophagus irregularis* = *Glomus intraradices*, *Funneliformis mosseae* = *Glomus mosseae*, and *Funneliformis coronatum*) applied plants at high temperatures.

In the maize plants treated with mycorrhizae under low-temperature stress, there was an increase in CAT activity in all the groups treated with mycorrhizae at 5 °C, 10 °C, and 15 °C compared to the M0 group (Fig. 5D). As indicated by the findings of *Zhu, Song & Xu (2010)*, there is a high level of CAT activity in plants inoculated with mycorrhiza under low-temperature conditions. In maize plants treated with mycorrhiza high temperature stress, high CAT activity was determined in plants treated with mycorrhiza species mixtures of M1 and M4 at 35 °C and M3 at 45 °C, compared with the M0 group. An earlier study indicated that there is an increase in CAT activity in mycorrhizal plants when they are exposed to high temperatures (*Jian et al., 2025*). This increase in CAT activity has been observed in maize plants treated with mycorrhiza.

## HSP70 and HSP90 gene expressions

Heat shock proteins (HSPs) are stress-induced chaperone protein complexes. Heat shock proteins range in size from 20 to 100 kDa and regularly interact with other proteins to perform their chaperone functions. The protective mechanism and repair ability of HSPs enable plants to maintain metabolism during periods of stress (*Singh, Gupta & Prasad, 2019*). HSP70 proteins are molecular chaperones that are powered by ATP. They have an ATPase domain at the N-terminus and a peptide-binding domain at the C-terminus. HSP70 is an evolutionarily conserved protein family that performs protein biogenesis, protects organisms during stress, prevents protein aggregation, assists in protein translocation, and performs multiple cellular functions. As the primary molecular chaperone protecting the cell, HSP70 plays a crucial role in protecting against various stress factors and restoring cellular homeostasis (*Sung, Kaplan & Guy, 2001*). It has been observed that the expression of HSP70 in maize plants changes according to high and low temperature stress and their combination when mycorrhiza is applied (Fig. 6A).

HSP90 proteins are found in most organisms and contribute significantly to cell homeostasis by playing a role in protein synthesis, stabilization, maturation, and activation. HSP90 proteins represent the most abundant proteins found in the cytosol, and they interact with a variety of other proteins, including calmodulin, actin, tubulin, kinases, receptor proteins, and chaperone groups in response to various stress factors. This interaction contributes to the cell's defense mechanism (*Xu et al., 2012*). The application of mycorrhiza resulted in alterations in HSP90 expression in maize plants under conditions of high and low temperature stress. These alterations were particularly evident at high temperatures (Fig. 6B). The application of the M2 at 10 °C and the M3 at 45 °C resulted in the greatest increase in both HSP70 and HSP90 gene expression.

In this study, the accumulation of the HSP70 and HSP90 proteins, which are considered as highly sensitive and significant markers for early warning at high and low temperatures, has been observed to increase under symbiotic conditions with mycorrhiza. This finding suggests that the plant's defense mechanism is operational. It can be posited that plants living in symbiosis with mycorrhiza exhibit enhanced protein metabolism and stronger defense mechanisms, particularly at temperatures where excessive expression is observed. A previous study indicated that the induction of HSPs by potential microorganisms is a functional genomic approach for determining heat stress tolerance during plant production (*Mitra, Djebaili & Pellegrini, 2021*). Specifically, high-temperature stress has been shown to activate the expression of HSPs, and heat shock transcription factors (HSFs) have been observed to rapidly bind to heat shock elements (HSEs) on HSP promoters, thereby activating the expression of downstream genes. This process has been demonstrated to enhance the tolerance of plants to high-temperature stress (*Scharf et al., 2012*; *Jian et al., 2024*). In the presence of heat stress, *Tian et al. (2023)* reported that an AMF (*Diversispora versiformis*) can enhance the heat stress tolerance of cucumber plants by inducing the expression levels of the CsHsp70 genes. Research has demonstrated that HSPs are induced in response to cold stress because of their functions in protecting membranes and preventing protein aggregation (*Timperio, Egidi & Zolla, 2008*). A study on sorghum revealed that the expression levels of certain SiHsp genes increased significantly in response to low-temperature stress (*Singh et al., 2016*). In a separate study on switchgrass, researchers ascertained that the expression of certain PvHsp20 genes increased significantly in response to low temperatures (*Yan et al., 2017*). *Tak (2023)* reported no significant relationship between HSP activation, one of the basic mechanisms that respond to heat stress, and AMF inoculation. Furthermore, it was emphasized that certain studies have demonstrated inconsistent outcomes concerning the regulation of specific genes involved in metabolic processes. These discrepancies may be attributed to a deficiency in comprehension of the functions of unique genes and proteins at the individual level. Consequently, the current study contributes to reducing inconsistencies in research results by investigating the genotypic compatibility between AMF and maize plants and understanding the effects of regulating HSP genes that provide plants with temperature stress tolerance.

## Correlations between the examined parameters and gene expressions

In addition to their role in stress-induced responses, HSP90s have been identified as critical components of the plant development process under normal physiological conditions (*Li, Dong & Qin, 2024*). *Xu et al. (2023)* examined the relationship between the HSP90.6 gene and carbon and nitrogen metabolism, finding that mutation of this gene caused defects in nitrogen recycling. They also reported that HSP90.6 interacts with components that control carbon and nitrogen metabolism. The present study hypothesizes that the observed negative correlation between plant N concentration and HSP90 gene expression can be attributed to this relationship. *Wang et al. (2014)* conducted a study on high-temperature stress in bentgrass and found that adequate N fertilization triggered the production of HSPs. However, it was also noted that HSP accumulation was lower at

higher N levels because of increased shoot growth. The induction of HSPs was reported to be a consequence of secondary effects of N. The increased HSP90 expression during high-temperature stress, as well as the increased N availability through mycorrhizal applications despite decreasing N concentrations, may have enhanced HSP90 induction. *Lucini & Bernardo (2015)* indicated that HSP70 and HSP90 levels increased in response to Zn under conditions of Zn stress. This phenomenon can be attributed to the increased expression of HSP90 in response to high temperatures. Mycorrhizal inoculation has been shown to enhance Zn uptake, leading to high Zn concentrations in plants. The observed negative correlation between the Zn concentration and heat stress can be explained by the decreased Zn concentrations resulting from heat stress.

In response to stressful conditions, HSPs utilize ROS as signaling molecules, thus preventing protein aggregation. *Yu et al. (2023)* reported that HSP90 affects antioxidant enzyme activity, indicating that SOD, POD, and CAT activities were higher in transgenic lines than in wild-type lines in the presence of salt. This enhanced tolerance to abiotic stress in transgenic tobacco is the result of the ROS-scavenging capacity of ZmHSP90 by inducing antioxidant enzymes. In our previous study, we found that HSP90 expression was positively and significantly correlated with SOD, APX, and CAT activity and negatively and significantly correlated with proline content under drought stress in maize (*Eskikoy & Kutlu, 2024*). In the present study, HSP90 was found to be positively correlated with MDA content, proline content, and GR enzyme activities. While increased MDA levels are associated with stress-induced damage, they can also function as a stress signaling molecule in plants, activating HSP genes (*Morales & Munné-Bosch, 2019*).

## CONCLUSIONS

The present study investigated the effects of different mixed AMF culture on the concentrations of nutrients, changes in physiological parameters, and gene expression of HSP70 and HSP90 in maize plants under low and high temperature conditions. Although mycorrhizal colonization decreased under heat stress, the highest colonization was observed in the M4 culture. However, M2 colonization was better preserved at low temperatures, and M1 colonization was better preserved at high temperatures. The dry weight of the plants decreased with increasing low and high temperature stresses, while the application of mixed AMF cultures increased the dry weight of the plants. Under all the temperature conditions, the M3 AMF culture resulted in the highest dry weight. However, the least dry matter loss occurred in the M2 application at 5 °C and in the M1 application at 45 °C. The concentrations of nutrients present vary depending on the AMF culture used under each temperature condition. Regardless of species and mixtures, mycorrhizal inoculation has been shown to enhance nutrient uptake. Applications that do not fall below the critical level of the relevant nutrient are recommended.

When physiological and molecular findings are evaluated in conjunction, HSP70 and HSP90 expressions reach their maximum at 10 °C, 40 °C, and 45 °C in all the mycorrhizal treatments. However, M2 symbiosis at 10 °C and M3 symbiosis at 45 °C reached the highest expression values. Furthermore, enzyme activities decreased at 5 °C, despite an

initial increase up to 10 °C. This finding suggests that the defense mechanism associated with HSPs and antioxidant enzymes in this maize plant may be reduced after 10 °C. The maize plant may have greater tolerance to high temperatures. However, antioxidative defense mechanisms vary depending on the differences in the responses of mycorrhizal species to temperature changes. While CAT, APX, and GR activities and proline accumulation were found to be high at low temperatures (10 °C) in plants treated with M3 mixed AMF culture, GR activity and proline accumulation were not maintained at high (45 °C) temperatures. However, SOD and GR activity persisted at higher temperatures in plants treated with M1 mixed AMF culture.

In conclusion, it was determined that low and high temperature treatments in maize caused significant changes in mycorrhizal symbiosis based on the parameters examined, and that these changes occurred at different levels depending on temperature changes and differences between mycorrhizal species. At 10 °C, in maize plants inoculated with M2 with the highest HSP70 and HSP90 gene expression, SOD activity and colonization rate; in maize plants inoculated with M3, the highest CAT, APX, GR activity, proline and dry matter accumulation, and nutrient uptake above critical levels could be effective in providing low-temperature tolerance. At 45 °C, maize plants inoculated with M3 may have most tolerant to high temperature with dry matter accumulation, along with elevated HSP70 and HSP90 gene expression, CAT and APX enzyme activity, and N and K uptake. In maize plants inoculated with M1 at 45 °C, the highest levels of SOD and GR activity, P and Mg uptake along with preserved colonization, may be effective in providing the highest tolerance. It has been demonstrated that maize plants can be grown safely under low temperature stress conditions by inoculating with M2 (*Glomus intraradices, Glomus constrictum*, and *Glomus microcarpum*), under high temperature stress conditions by inoculating with M1 (*Glomus mosseae* and *Glomus intraradices*), and under both conditions by inoculating with M3 (*Gigaspora sp., Glomus constrictum*, and *Glomus fasciculatum*). The results of this study not only reveal the metabolic effects of low- and high-temperature stress on mycorrhizal maize plants but also provide recommendations for appropriate mycorrhizal species mixtures to be applied to plants.

## ACKNOWLEDGEMENTS

This manuscript is based on Vedia Turudu's master's thesis. The authors would like to thank Aysel Bars Orak for their assistance with supplies M1, M2, and M3 mycorrhiza cultures and to MST Biotechnology Laboratory for their technical support in conducting RT-qPCR.

### Funding

This research was supported by the Scientific Research Projects Commission of Eskisehir Osmangazi University, grant number FYL-2022-2288. The funders had no role in study design, data collection and analysis, decision to publish, or preparation of the manuscript.

## Grant Disclosures

The following grant information was disclosed by the authors:

Scientific Research Projects Commission of Eskisehir Osmangazi University: FYL-2022-2288.

## Competing Interests

Imren Kutlu is an Academic Editor for PeerJ.

## Author Contributions

- Vedia Turudu conceived and designed the experiments, performed the experiments, prepared figures and/or tables, and approved the final draft.
- Imren Kutlu conceived and designed the experiments, performed the experiments, analyzed the data, prepared figures and/or tables, authored or reviewed drafts of the article, and approved the final draft.
- Nurdilek Gulmezoglu conceived and designed the experiments, performed the experiments, authored or reviewed drafts of the article, and approved the final draft.

## Data Availability

The raw data is available in the Supplemental File.

## Supplemental Information

Supplemental information for this article can be found online at http://dx.doi.org/10.7717/peerj.20419#supplemental-information.

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
