# Peer review of "Biochemical and molecular responses of maize to low and high temperatures in symbiosis with mixed arbuscular mycorrhizal fungi cultures"

_PeerJ, doi:10.7717/peerj.20419_

## Round 0.1 · original submission · Major Revisions

· Academic Editor

Major Revisions

Dear Dr. Kutlu, I ask you to make changes to the article in accordance with the reviewers' comments.

**Language Note:** PeerJ staff have identified that the English language needs to be improved. When you prepare your next revision, please either (i) have a colleague who is proficient in English and familiar with the subject matter review your manuscript, or (ii) contact a professional editing service to review your manuscript. PeerJ can provide language editing services - you can contact us at [email protected] for pricing (be sure to provide your manuscript number and title). – PeerJ Staff

Reviewer 1 ·

Basic reporting

I have carefully reviewed the manuscript entitled “Biochemical and molecular responses of maize to low and high temperatures in symbiosis with mixed arbuscular mycorrhizal fungi cultures” submitted by Turudu et al. to PeerJ. This study assessed the effect of temperature change and AMF symbiosis on mineral element concentrations, physiological parameters, and gene expression of heat shock proteins in maize plants. Research background was described clearly and this was an interest study with actual production significance at a certain degree.

Experimental design

The experimental design, statistical methods, and testing and calculation procedures were correct. The manuscript was well-written and well-structured. This study provided some useful data and these data could advance the understanding of resistance mechanism of maize to heat stress.

Validity of the findings

I suggest accepting this manuscript after minor revision. Some comments are as the following:
1. Line 165-170. The heat stress only lasted for 24 hours, which is significantly inconsistent with actual agricultural production.
2. Supplementing vertical axis title and statistical analysis results in Figure 1.
3. Give a clear conclusion at the end of Abstract section, namely, which temperature and AMF are suitable for maize production under potential heat stress.
4. This study determined many items and provided many data. Please try to draw a mechanism diagram representing how maize resists heat stress based on the results from this study.

Reviewer 2 ·

Basic reporting

In general, the manuscript is well-organized and clearly structured. The required data and figures are presented.
English does not affect understanding, and seems to be acceptable.
Literature review and reference style are correct. However, I would recommend the authors to avoid quotations citing more than two references like this on line 73 ((Zhu et al. 2010; 2015; Liu et al. 2016a; Devi et al. 2019; Li et al. 2020). I feel that such citations must be split.
Introduction lacks clarity in main hypothesis and goals statements.

Experimental design

Materials and Methods section lacks "Location and duration of the experiment" sub-section to clarify where and when the experiment was conducted.
The experimental design itself is robust, and it is clearly explained step-by-step.
The study could be replicated in case of need.
The authors claim about standard error bars on the plots. However, not all the plots have error bars (e.g., Figure 1 has none).

Validity of the findings

The findings are well-presented and have reasonable substantiation from the authors.
However, it is quite debatable that the correlation analysis "HSP90 gene expression
revealed negative and significant correlations with N, K, Mg, Fe, Zn, Mn, and Cu concentrations,
as well as positive and significant correlations with MDA, proline content, and GR activity" (lines 410-412).
Having examined Table 1, it was found out that the maximum |R| = 0.62, and this value is far from being strong correlation. It might be statistically significant, but these values claim about just moderate relation between the studied indicators. Therefore, I feel that the results of correlation analysis require further deeper examination and explanation.

Additional comments

I feel that the study is scientifically sound and well-performed.
The manuscript deserves to be published after minor revision.

Reviewer 3 ·

Basic reporting

General Comments:
This manuscript investigates the important and timely topic of how different mixed cultures of arbuscular mycorrhizal fungi (AMF) influence the biochemical and molecular responses of maize to both low and high temperature stress. The authors have conducted a comprehensive experiment observing various physiological and molecular parameters. The study provides valuable insights into the potential of AMF to mitigate temperature stress in crops. However, significant improvements in the manuscript's content, presentation, and clarity are required before it can be considered for publication. The specific points needing attention are detailed below.
Major Comments:
• Comment 1 (Rationale for AMF Selection): The manuscript's introduction and methods lack a clear scientific rationale for selecting the specific mixed AMF cultures used in this study. The authors should provide background information on these cultures. Why were these particular combinations chosen? Is there existing evidence to suggest they might be particularly effective in conferring tolerance to temperature extremes? Please elaborate on the composition and origin of each mixed culture (M1, M2, M3, M4) to justify their inclusion in this experimental design.
• Comment 2 (Graphical Summary): The results are complex, and a summary figure would greatly enhance the reader's comprehension of the key findings. I strongly recommend the authors create a graphical abstract or a concluding summary figure. This figure should provide a schematic overview of the proposed mechanisms by which AMF symbiosis modulates maize's response to cold and heat stress, comparing these effects to the normal temperature condition. This visual aid would be highly beneficial for readers to follow and understand the main outcomes of this research.
• Comment 3 (Discussion of AMF Strain Specificity): The discussion section notes that different AMF mixtures had varying effects under different temperature conditions. This is an interesting finding that requires deeper exploration. The authors should expand the discussion to consider the specific fungal species or strains composing each mixture. Please discuss any existing literature that reports on the differential efficiency of specific AMF species/strains in improving plant tolerance to abiotic stresses, particularly heat or cold. Linking the observed results to the known characteristics of the fungi within each mixture would significantly strengthen the manuscript's conclusions.
• Comment 4 (Introduction and Link to Plant Defense): The authors focus on Heat Shock Proteins (HSP70 and HSP90) as stress indicators. However, these proteins are also integral components of plant defense signaling, often linked to pathways involving Jasmonic Acid (JA) and Salicylic Acid (SA). To provide a more complete context, the introduction should be expanded to include information on plant defense responses under abiotic stress. Furthermore, given that the study measures antioxidant activity (e.g., SOD), the introduction should also establish a stronger connection between temperature stress, oxidative damage, and the importance of antioxidant enzyme systems in plant tolerance mechanisms.
Minor/Specific Comments:
• Figure 1 & 3: The y-axis label is missing in Figure 1. Please add it. Additionally, for both Figure 1 and Figure 3, the details of the statistical analysis performed should be clearly indicated in the figure captions or directly on the figures themselves (e.g., by using letters to denote significant differences between groups).
• Line 497: There is a typographical error. Please change "Thissuggests" to "This suggests".
• Line 553: The term “mixed AMF culture” is ambiguous here. Please clarify if this specifically refers to the M4 treatment or another combination.
• Line 557: There appears to be a stray equals sign. Please delete the "=" in the phrase "As indicated =".

Experimental design

no comment

Validity of the findings

no comment

Additional comments

no comment

Annotated reviews are not available for download in order to protect the identity of reviewers who chose to remain anonymous.

---

## Round 0.2 · Major Revisions

· Academic Editor

Major Revisions

Dear Dr. Kutlu, I request that you address Reviewer 4's key comments before this article is accepted for publication.

Reviewer 1 ·

Basic reporting

no comment

Experimental design

no comment

Validity of the findings

no comment

Additional comments

I have reviewed this resubmitted manuscript. Some questions raised previously have been addressed. Regarding the remaining unsolved questions, authors also provided sincere and reasonable explanations. Therefore, I suggest accepting this manuscript for publication.

Reviewer 3 ·

Basic reporting

The manuscript is clearly written, well-structured, and supported with relevant literature. Figures and tables are clear and appropriately presented.

Experimental design

The methodology is described in sufficient detail to allow reproducibility.

Validity of the findings

The results are well-analyzed and convincingly support the conclusions.

Additional comments

The authors have thoroughly addressed all of my previous comments and significantly improved the manuscript entitled “Biochemical and molecular responses of maize to low and high temperatures in symbiosis with mixed arbuscular mycorrhizal fungi cultures.”

I find the current version satisfactory, and I have no further concerns. In my opinion, the manuscript is now suitable for publication in PeerJ.

Reviewer 4 ·

Basic reporting

The article is highly interesting and timely in the context of climate change. Global climate change is leading to significant alterations in agricultural cultivation technologies, especially for crops of worldwide importance. Maize is one such crop, and this research could subsequently influence the scheduling of technological operations in its cultivation.

Experimental design

However, despite the relevance of the topic, the following remarks should be considered:
1. In the "Results" section, the authors did not devote any attention to the development of the root system and the impact of the studied temperatures on its development. This is particularly important given that microbial flora development occurs on the root system. Furthermore, there is no mention of root system development in the "3-liter plastic pots" – was this volume sufficient for proper root development? Could this have influenced the development of microbial colonies?
2. The authors did not present the results of statistical analysis, although it is mentioned in the methodology description (lines 316-323). It is unknown how to verify the data obtained.
3. Given that the maize root system is deep-penetrating, the authors need to explain why soil samples were taken only from the 0-30 cm layer (line 151).
4. The authors indicated the use of "3-liter plastic pots," but to clearly understand the nutrition area per plant, the specific dimensions of the pots should be provided (lines 175-176).
5. When describing the experiment, the authors must first explicitly state which specific research method was applied, and only then characterize it and describe the experimental conditions (lines 174-179).
6. The authors should provide an explanation as to why they used the number of days (line 203) when describing the growth and development phases of the crop, not the international classification of the ВВСН. This does not give a clear idea of how the crop develops, how many leaves it has, what kind of root system it has, etc.
7. In the description within the subsection "Experimental setup, mycorrhiza applications, and plant cultivation," the authors did not indicate the sowing depth of the maize seeds (lines 189-195).
8. The rationale for selecting the specific temperature level mentioned in lines 206-207 needs to be explained.
9. When describing the results, the authors should state whether colonization of the maize root system by indigenous microflora occurred in the control treatment (lines 326-334).
10. It is suggested that the conclusions be improved through more precise detailing and the removal of general statements. The conclusions should be clear and logical, and their overall length should be reduced.
11. In the "Materials & Methods" section, the authors must provide references for the geographical locations of the study sites in lines 151-152.
12. In lines 319-320, the authors did not indicate which specific figures present the mentioned values.
13. In line 180, the authors failed to specify the type of alcohol used.
14. In line 217, the authors did not specify the type of alcohol used.
15. The authors need to review the accuracy of the "References", particularly the listing of all authors (e.g., lines 864-867, 882-885, 889-891, 895, 898, 905, 973, 979, 990, 1023-1024 (reference is missing), 1025, 1060, 1069, 1082, 1093).

Validity of the findings

Language and Data: The English language is comprehensible and, in my opinion, does not require improvement.

I thank the authors for providing the raw data.
The authors provided good illustrative material, but some aspects are impossible to analyze due to the very small-scale figures.
The presented data are interesting but could be enhanced by: Aligning the timing of observations/treatments with the maize developmental phases (BBCH scale); Providing data on root system development; Improving the research methodology description by specifying the experimental design method and the parameters of the cultivation vessels.

Additional comments

I thank the authors for providing the primary research data. Furthermore, the manuscript is written in a clear, professional, and unambiguous style. However, there are remarks concerning the improvement of the research methodology, specification of crop development stages, provision of information regarding the root system, and issues with the statistical analysis (as noted above), which should be addressed prior to acceptance.

---

## Round 0.3 · accepted · Accept

· Academic Editor

Accept

Dear Dr. Kutlu, I congratulate you on the acceptance of this article for publication.

Reviewer 4 ·

Basic reporting

I have reviewed this resubmitted manuscript. Questions raised previously have been addressed. The authors also provided sincere and reasonable explanations. Therefore, I suggest accepting this manuscript for publication.

Experimental design

no comment

Validity of the findings

no comment

Additional comments

no comment